# Extracting Retinal Anatomy and Pathological Structure Using Multiscale Segmentation

**Lei Geng [1,2], Hengyi Che [1,2], Zhitao Xiao [1,2,]\*, and Yanbei Liu [1,2]**

1   School of Electronics and Information Engineering, Tianjin Polytechnic University, Tianjin 300387, China
2   Tianjin Key Laboratory of Optoelectronic Detection Technology and System, Tianjin Polytechnic University, Tianjin 300387, China
\*   Correspondence: xiaozhitao@tjpu.edu.cn

**Abstract:** Fundus image segmentation technology has always been an important tool in the medical imaging field. Recent studies have validated that deep learning techniques can effectively segment retinal anatomy and determine pathological structure in retinal fundus photographs. However, several groups of image segmentation methods used in medical imaging only provide a single retinopathic feature (e.g., roth spots and exudates). In this paper, we propose a more accurate and clinically oriented framework for the segmentation of fundus images from end-to-end input. We design a four-path multiscale input network structure that learns network features and finds overall characteristics via our network. Our network's structure is not limited by segmentation of single retinopathic features. Our method is suitable for exudates, roth spots, blood vessels, and optic discs segmentation. The structure has general applicability to many fundus models; therefore, we use our own dataset for training. In cooperation with hospitals and board-certified ophthalmologists, the proposed framework is validated on retinal images from large databases and can improve diagnostic performance compared to state-of-the-art methods that use smaller databases for training. The proposed framework detects blood vessels with an accuracy of 0.927, which is comparable to exudate accuracy (0.939) and roth spot accuracy (0.904), providing ophthalmologists with a practical diagnostic and a robust analytical tool.

**Keywords:** retinopathy feature; deep learning; classified database; image processing; multiscale segmentation

## 1. Introduction

Retinopathy is a symptom of many common diseases, such as diabetes, arteriosclerosis, and leukemia [1–3]. These diseases are known to easily spread to blood vessels in the retina [4]. Therefore, many patients with these afflictions are often screened and diagnosed by analyzing retinal fundus images. These retinal arteries and veins can also be analyzed to determine whether the patient has arteriosclerosis. There are several features, e.g., bleeding points and exudates, that can be used to determine diabetic retinopathy. Patients can often judge glaucoma via an optic disc detection method. Therefore, to accurately diagnose these diseases early, a quantitative method needs to be developed. In the past, analysis of the various parts of the retina played a decisive role in disease detection and prevention. Therefore, retinal image segmentation can provide doctors and experts with a highly efficient retinal disease diagnosis method that can also prevent disease. Deep learning-based image segmentation is the embodiment of artificial intelligence.

The segmentation of the fundus of the retina has long been the focus of experts around the world. There are many traditional state-of-the-art algorithms for segmenting the fundus of the eye, and these algorithms can be broadly divided into two categories: Unsupervised and supervised [5,6]. The image

segmentation methods in medical imaging includes superpixel segmentation methods [7,8], watershed segmentation methods [9,10], and active contour methods [11–13]. In 2016, Zhiqiang Tian et al. [7] proposed a superpixel-based 3D graph-cutting algorithm that segments prostate surfaces. Nguyen et al. [8] propose a multi-atlas fusion framework to automatically segment pelvic structure from 45 to 135-degree oblique X-ray radiographic images. In 2004, Yu-Len, Huang et al. [9] integrate the advantages of neural network (NN) classification and morphological watershed segmentation to extract precise contours of breast tumors from US images. Hassan Masoumi et al. [10] present a new automatic system for liver segmentation from abdominal MRI images. Marcin et al. [11,12] analyzed comparative research to assess the utility and performance of three active contour methods (ACMs) for segmenting the corpus callosum (CC) from magnetic resonance (MR) images of the brain. Zhao Y et al. [13] propose a new infinite active contour model that uses hybrid region information to approach segmentation of blood vessel. In recent years, deep learning algorithms have made key breakthroughs by finding shallow features from abstract deep features. Compared with traditional methods, deep learning uses a computer program to autonomously learn from large sets of imaging data. These methods often solve problems that can be solved by traditional algorithms at lower computational cost and with better efficiency. Many segmentation methods for fundus images have been widely reported. For example, Khalaf et al. [14] simplified a convolutional neural network (CNN) structure to distinguish between large and small blood vessels (and background) in fundal images. These methods further adjust images of different sizes via convolution kernels and can often obtain good segmentation results using a digital retinal image blood vessel extraction library [15]. Fu et al. [16] proposed a retinal image vessel segmentation method based on CNNs and a conditional random field (CRF). This method treats the blood vessel segmentation step as a boundary detection problem and uses a CNN to generate a segmentation probability map. Finally, these methods combine CRFs to obtain a binary diagnostic result. AlexNet [17] and VGGNet [18] methods were proposed for image segmentation next, but there are still many drawbacks. First, the image database is a limitation that renders label indexing inaccurate. Second, retinal images with large-area pathological features or lesions interfere with normal regions, which seriously affects the segmentation effect. Finally, the image quality of a small number of retinas is affected by the environment at the time of collection, resulting in low feature detail and background contrast. Manual labeling of datasets depends on an operator's technical experience, which means that the number of subjective factors is large, thus rendering the method inefficient [19].

By comparing various networks, we find that the PixelNet [20] model (based on the number of pixel points of a feature map) is the most effective segmentation finder; this is suitable for retinal fundus image segmentation. The model uses a VGG-16 convolution to extract convolution features and extracts the corresponding features from multiple convolutional feature maps for a single sampled pixel. This creates a hyper-column descriptor that can be inputted as a feature of an MLP multilayer perceptron. Finally, the output of the classification result is determined. In addition, the main idea of the network lies in the sampling strategy used during training, which can accelerate the training step. Pixel-based stratified sampling or sparse sampling is often a problem. However, even when we sample from very few pixels of a training image, we can nevertheless get good results. This work also describes how to assemble an image database to provide a broader training set. Then, by using the PixelNet network structure, we replaced the original VGG-16 with VGG-19 to increase the width of the network. We also adjusted the output of each layer of the network structure to be consistent. Therefore, we can adjust the number of feature maps generated by each layer. We also notice that we can maximize the size of the compression model to improve the test rate under the condition that model accuracy is guaranteed. Therefore, the multiscale segmentation framework is achieved using our two contributions:

1.  A task-specific network that is designed to compute the retinal anatomy and pathological structure of a fundus. We collected a large-scale retinal dataset composed of 1500 images from 282 proliferative diabetic retinopathy (PDR) and non-proliferative diabetic retinopathy (NPDR)

patients. PDR and NPDR are indicators for judging diabetic retinopathy grades. The difference between PDR and NPDR lies in blood vessel formation. We also know that PDR develops after NPDR. Therefore, we design an adaptive thresholding method to segment the FOV boundary from the background of the retinal fundus image. The design of the network can often support the boundary FOV segmentation and the automated segmentation training system, which is consistent with clinical requirements.

2. A multiscale network is proposed to achieve the final segmentation step, e.g., a four-input branch that enhances the integrity of segmentation and the ability to generalize the training model. The purpose does not neglect the integrity of the structure while training. The network is also error-tolerable, which means that the diagnostic accuracy can be guaranteed even when one of the branches is not accurate enough.

Therefore, the innovation of our method lies in the four input branches. The network structure is not limited to segmentation of single retinopathy features. Our method is suitable for exudates, roth spots, blood vessels, and optic discs segmentation, and our database has an adaptive thresholding method to segment the field of view (FOV) boundary from the background of the retinal fundus image.

In addition, we change the number of layers according to the complexity of the different features of the fundus to improve the method's efficiency. Finally, the network models are distributed to multiple GPUs for segmentation and validation, based on the test image set, 0 to achieve independent segmentation between the various structures in the images.

## 2. Methodology

In this section, we provide the details of our multi-segmentation framework that diagnoses diabetic retinopathy using clinical features, including a task-specific network that categorizes the structure of the fundus and a network that determines the final segmentation boundaries.

### 2.1. PixelNet

The network first uses a VGG-16 convolution to extract convolution features; it extracts the corresponding features from multiple convolutional feature maps for a single sampled pixel. A hyper-column descriptor is generated, and the inputs of the feature are sent to an MLP multilayer perceptron. Finally, the classification result is output by the system. The main idea of PixelNet is the sampling strategy during training, which speeds up the training. A pixel-based stratified sampling method and a sparse sampling method are proposed next. Thus, very few results can be obtained by sampling a small number of pixels from the training images. The network verification shows that the network model has good semantic segmentation, surface normal estimation, and edge detection. Next, we analyze the characteristics of the network.

The network uses a VGG16 CNN classifier [21]. An improvement of the VGG16 over AlexNet replaces several larger convolution kernels ($11 \times 11$, $5 \times 5$) in AlexNet with several consecutive $3 \times 3$ convolution kernels. For a given receptive field (the local size of the input image associated with the output), the stacked small convolution kernel method is superior to the large convolution kernel because the multi-layer nonlinear layer can increase the network depth to ensure accurate learning and detect more complex patterns at a lower cost with less parameters. The characteristics of the network includes the use of 13 convolution layers and 3 fully connected layers to form the VGG16. The convolution layer is now described as:

CONV = $3 \times 3$ filter, step = 1, padding = "same" (The input image size is the same as the output image size).

Next, we keep the shape of the modified image, after each convolution set and use the MAX_POOL command to down sample and reduce the image size to speed up the calculation speed:

MAX_POOL = $2 \times 2$, s = 2.

The MLP multilayer perceptron [22] collects the features, extracted from the VGG16, and finally outputs the classification result. First the network loads the image into the convolutional neural

network and extracts hyper-column descriptor from multiple volumes. Then, it feeds the descriptor to the MLP. Finally, the segmentation result from the last layer of the MLP is generated.

Next, we focus on the accelerating stochastic gradient (SGD) descent method. We refer the reader to [23] for an excellent introduction. Though naturally a sequential algorithm that processes one data example at a time, recent SGD work focuses on mini-batch methods that can exploit parallelism in GPU architectures or clusters [24]. One general theme is an efficient online approximation of second-order methods [25], which can model correlations between input features. Batch normalization [26] computes correlation statistics between samples in batch, producing noticeable improvements in convergence speed. Our work builds upon similar insights and directly uses convolutional networks without explicit second-order statistics.

### 2.2. Multiscale Segmentation Network

As in past work, our architecture makes full use of multiscale convolutional features, which we write as a hyper-column descriptor $h(p)$ characterized by

$$h(p) = [c_1(p), c_2(p), \ldots, c_M(p)]. \tag{1}$$

Then we learn a nonlinear predictor, $g[h(p)]$, which is implemented as a multi-layer perception (MLP) defined over hyper-column features. We use an MLP with ReLU activation functions, which can be implemented as a series of "fully-connected" layers. The last layer must have size K, e.g., the class labels or real valued outputs being predicted. This is based on the VGG19 network level but is really deeper than VGG16; thus, to increase the network width, VGG19 is used instead of VGG16 structure. PixelNet is a single-scale input that uses $224 \times 224$-pixel images for training. This brings great inconvenience. Some large size images are often processed in blocks, while the final result is stitched and integrated; this results in a GPU load that has been increased several times, greatly reducing efficiency. The overall outline of the image cannot be learned since splicing can also lead to edge errors. Inspired by the Unet multiscale network architecture, we apply our improved PixelNet network to multiscale input, e.g., inputting $224 \times 224$, $448 \times 448$, $896 \times 896$ and complete images. In this way, the size of the image and the framework output is no longer limited to $224 \times 224$. Thus, the Unet network structure becomes a multiscale method. Next, we propose the network structure shown in Figure 1 for multiscale input (blood vessels are taken as an example), and the final output of the blood vessel results are more integrated.

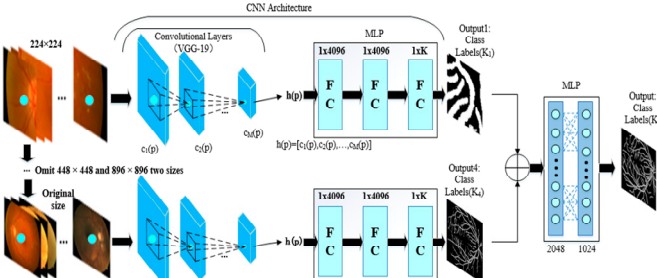

**Figure 1.** The structure of multi-seg convolutional neural network (CNN).

In this part, the four-branch network extracts features that are independent, and the branch parameters are not shared. Therefore, at the beginning of the data input, the four branches train the original data, while the features of the last convolution layer of each channel are extracted after the four branches are convolved. The vectors are concatenated to make the feature information richer. The feature vectors of the four channels are spliced at a ratio of 1:1:1:1. The result is output to the next fully connected layer so that it fully integrates the respective characteristics of the four-branch network, taking into account the pre-training model. The fully connected layers predict the label of a

single pixel in our method. The network pre-training structure of each channel is consistent with the PixelNet network, except for the patch size parameter.

In the era of big data, deep learning algorithms will replace most traditional algorithms. Our network structure consists of four separate branches. The details and integrity of the target are enhanced compared to a single branch network. However, we have to spend more training time and trainable parameters to deal with more fully connected layers with four-branch structure. In other words, the chance of overfitting is increased. We surmise the overfitting from generalization error.

Through the design of the network structure, a four-branch network is used to train the model in a more general way. Finally, a better segmentation effect is achieved by using this model to segment a fundus image.

## 3. Experiments

In this chapter we will mainly introduce our segmentation results and compare it with different segmentation networks. The proposed method is implemented on GPU (GeForce GTX 1080 Ti) and CPU (3.4GHz Intel core), caffe framework is used as well.

### 3.1. Data Preparation

We have refined and built our own optimized database based on the characteristics of the public database.

### 3.1.1. Public Database

The existing retinal fundus database has a complete DRIVE library, DIARETDB0 library, DIARETDB1 library, MESSIDOR database, e-Optha EX and e-Optha MA library, and a STARE library. Each library has a blood vessel labeling database. Basically, features other than vascular features are missing in these databases.

The DRIVE database [27] consists of 40 color retinal fundus images captured by a Canon non-mydriasis 3CCD camera with a resolution of 768 × 584, 45° FOV, and 8-bit channels per color. The database contains 33 healthy states and seven pathological retinal fundus images. The set of images is divided into a training set and a test set, both of which contain 20 images. The database provides true annotation of blood vessels for the test set. However, the data in the dataset are limited, and the image resolution is single, which is not ideal for additional fundus image testing. The DIARETDB0 and DIARETDB1 databases [28] contain 130 color retinal fundus images. The 130 retinal images available in the database include 110 markers depicting diabetic retinopathy (e.g., hard exudate, soft exudate, micro aneurysm, or hemorrhages). The images consist of 20 normal retina images. The DIARETDB1 database consists of 89 colored retinal fundus images. The 84 retinal images available in the database contain mild signs of diabetic retinopathy, including exudates, and the remaining five are healthy retinal images. However, these labels only contain relevant areas of the lesion in the retinal image.

The MESSIDOR database [29] consists of 1200 retinal fundus images taken with a Topcon TRC NW6 non-mydriasis fundus camera with 1440 × 960, 2240 × 1488 and 2304 × 1536 pixels. This database provides diagnostic results for various retinal images, but the database does not provide a reference standard for segmentation of fundus retinopathy.

The e-Optha EX and e-Optha MA databases are described in [30]. The e-Optha EX database consists of 82 retinal fundus images with 35 healthy retinal images and 47 pathological retinal images with pathological retinal images and mild exudate composition associated with moderate and severe diabetic retinopathy. The e-Optha MA database consists of 233 healthy retinal fundus images and 148 images with MA or small hemorrhage. These annotations are useful for accurately assessing the efficiency of exudate and MA segmentation methods. Unfortunately, there is no indication of common features like the cup optic disc or macula.

The STARE database [31] consists of 400 color retinal fundus photographs with a resolution of 605 × 700. This database provides diagnostic results for diabetic retinopathy based on a single retinal

fundus image, but it does not contain areas of various retinopathy structures, such as exudates and roth spots. Moreover, after careful comparison of the data, database labeling is very different from the original image, while edge processing has insufficient detail, which has a serious impact on the training results. After a detailed comparison and fitting, only 20 groups of retinal data have reference values.

In summary, the open source benchmark database has the following limitations: (1) contains a limited number of retinal images and the pixel resolution is generally low; (2) contains only clear contrast and original retinal images; (3) does not contain ground-truth information, e.g., labeling of all the retina's features; (4) the annotations are significantly different from the original ones and some have no annotation detail.

### 3.1.2. Optimized Database

Our comprehensive database contains 1500 retinal fundus images of 283 patients. There are 444 1444 × 1444 images and 940 2124 × 2056 images. The images are from individuals ranging from 25–83 years old for men and 24–75 years old for women. Additionally, each pathological grade is approved by the hospital ophthalmologist. Most of the retinal fundus images are longitudinal, i.e., data contain images for a single patient over a period of time, which makes it easier to analyze the condition later. The structure of the database is shown in Figure 2. In the experiment, we use optimized database to train and test the network segmentation model.

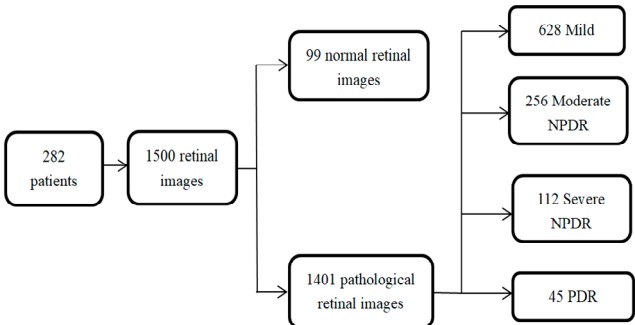

**Figure 2.** Distribution of clinically acquired database images among various stages of diabetic retinopathy.

Our database has improved in nature. It is not only a simple expansion of the number of open source databases but also contributes to the simple classification of disease level. The most important goal is to solve the problem of poor quality of some fundus photos, such as retinal images containing dark areas, so that the fundus image can be separated from the background image, as shown in the third column of Figure 1. Therefore, the first step is to segment the retinal image from the background. We grouped image pixels with similar properties by using a *k*-means clustering method [32] to reduce the number of different colors in the retinal fundus image and perform an adaptive thresholding method to segment the FOV boundary from the background of the retinal fundus image. Since the goal is to distinguish the FOV from the background, the threshold must be greater than the intensity value of the background pixel. However, we need to choose a larger threshold parameter value. The observation is compared by changing the threshold from 0 (corresponding to black) to 30. Initially, FOV segmentation improved as the threshold increased, but only after a certain number of trials (23). Our results show that there was no significant improvement in the segmentation. The FOV boundaries become distorted. This is due to the inclusion of pixels corresponding to the inner region of the FOV. Therefore, we re-set the threshold to 23. Some retinal images and corresponding segmented FOVs are given in Figure 3. Figure 3a–c shows an original retinal fundus image with a dark background area, a full field-of-view circle and a truncated circle. We see that that the proposed method is capable of segmenting the FOV without being affected by the dark areas of the fundus image, as shown in Figure 3d–f. This can also solve the problem of false detection of FOV edges, as shown in Figure 4.

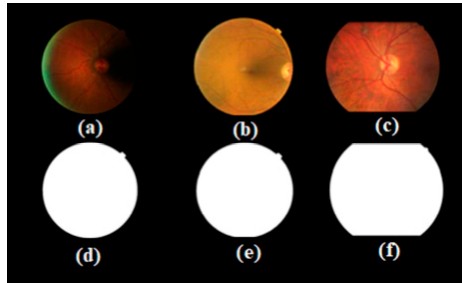

**Figure 3.** Original retinal image (**a**–**c**) and corresponding segmented FOV (**d**–**f**).

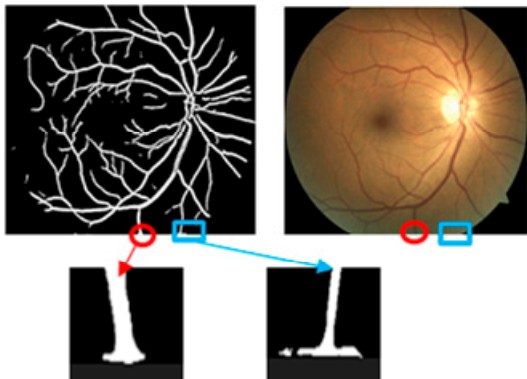

**Figure 4.** Edge misdetection after split FOV.

We accurately labelled the blood vessels, bleeding points, exudates, optic discs, and macula in the FOV. Hospital clinical ophthalmologists continuously calibrated the labeled results in real time. Eventually all results were corrected by the fundus specialist to achieve the most accurate labeling results.

### 3.2. Segmentation Criteria

To more objectively compare the similarities and differences between the segmentation results and the artificial calibration results, four quantitative statistical indicators are introduced: (1) true positives (TP) refers to pixels that are actually blood vessels that are accurately recognized by the model as blood vessels; (2) false negatives (FN) refer to pixels that are actually blood vessels but are recognized as non-vascular by the model; (3) true negatives (TN) refer to pixels that are actually non-vascular and are accurately identified by the model as non-vascular; (4) false positives (FP) refer to pixels that are actually non-vascular but are recognized by the model as blood vessels. The parameters of the network trained by the parameters sensitivity (Se), specificity (Sp), accuracy (Acc), and other parameters are evaluated via the expression:

$$Se = TP/(TP + FN), \tag{2}$$

$$Sp = TN / (TN + FP), \tag{3}$$

$$Acc = (TP + TN)/(TP + FP + TN + FN), \tag{4}$$

the receiver operating characteristic (ROC) curve is drawn according to the relationship between Se and 1-Sp. The area under the ROC curve (area under curve, AUC) reflects the performance of the segmentation method. The AUC is 1. The classifier is a perfect classifier. We first validate the performance of our proposed method via a grading method that uses $K_1$, $K_2$, $K_3$, $K_4$ and binary segmentation. The detection of diabetic eye disease plays a critical role in clinical diagnosis, where the false positive and false negatives need to be avoided. Thus, we validate the performance using multiple

evaluation protocols, including accuracy, sensitivity, and specificity. Table 1 records the performance of all testing images using the $K_1$, $K_2$, $K_3$, $K_4$ and binary segmentation evaluation protocols.

**Table 1.** Validation of the performance accuracy.

|  | $K_1$ | $K_2$ | $K_3$ | $K_4$ | **Binary** |
|---|---|---|---|---|---|
| Accuracy | 0.9551 | 0.9433 | 0.9356 | 0.9233 | 0.9271 |
| Sensitivity | 0.9432 | 0.9315 | 0.9235 | 0.9126 | 0.9182 |
| Specificity | 0.9491 | 0.9345 | 0.9239 | 0.9138 | 0.9158 |

From the Table 1 we know that the higher the accuracy of the input images (image size increases) in the branch, the lower the accuracy of the image. This is because the input image pixel value increases too fast. Figure 5 provides the ROC curve of our multi-seg-CNN.

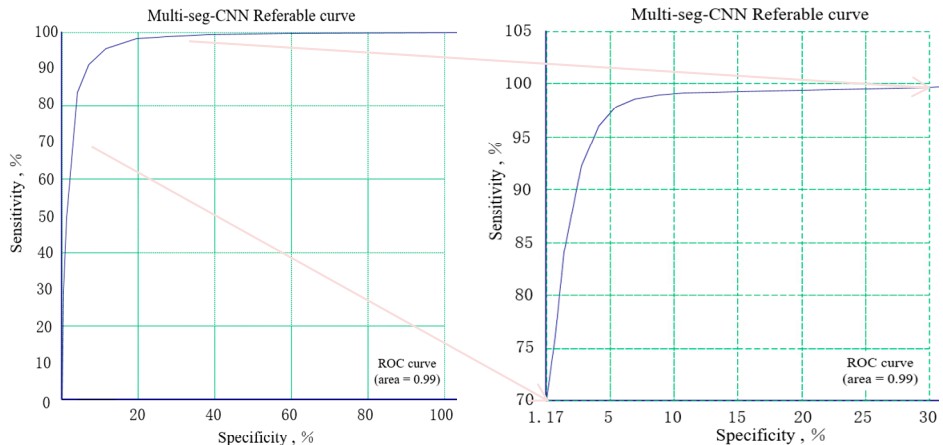

**Figure 5.** The receiver operating characteristic (ROC) curve of our multi-seg-CNN.

Additionally, as shown in Figure 5, we provide the ROC curve of our multi-seg-CNN. According to Figure 5, our method can achieve excellent performance during the automated diagnosis step and can meet the requirements of retinal anatomy and pathological structure.

### 3.3. Training

The various structures of the retina are different. Each structure can be divided into two categories according to our network structure. One type is called the concentrated structures, e.g., optic discs and the macula. The other types are called the overall structure, e.g., blood vessels, exudates, and roth spots. The characteristics of the concentrated structure are that the retinal structure of the retina usually appears only in a certain region of the image, and the proportion is smaller than the overall image, thus the structural integrity is lower. The characteristics of the overall structure are that the retinal structure generally appears in various places of the image, and the proportion is larger than that of the whole image.

In summary, we use two data input methods. For the concentrated structure, we perform a preprocessing step, as shown in Figure 6. We extract the concentrated structure of the complete image, such as (a) and (b) in Figure 6. The segmented structure is extracted more clearly. Then the concentrated structure is selected as the input. The purpose of this step is to adapt to the network. We try to increase the proportion of the centralized structure in the whole image. This structure is used to improve the speed because it is relatively simple. Therefore, we use a single scale as the input, e.g., scale of 224 × 224. For the overall structure, we use the multiscale input network shown in Figure 1. The input is 224 × 224, 448 × 448, 896 × 896, and complete image. Each image inputs the details by not

ignoring the integrity of the overall structure, thus deriving the weight based on the back-propagation algorithm [33]. The block diagram of the training parameter algorithm is shown in Figure 7. There is no need to distinguish between the two structures during training.

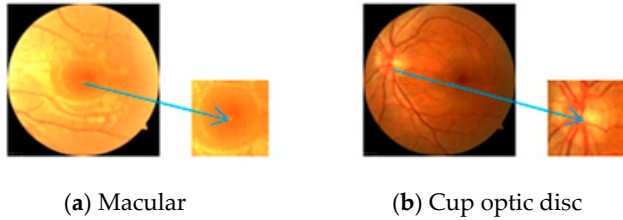

(**a**) Macular        (**b**) Cup optic disc

**Figure 6.** Preprocessing of concentrated data.

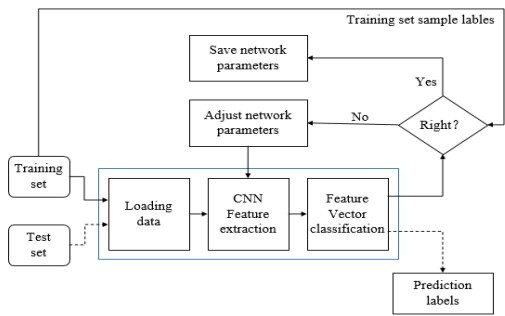

**Figure 7.** Training flowchart.

In our method, each branch has a loss function. We denote the *Loss* $K_1$, *Loss* $K_2$, *Loss* $K_3$, *Loss* $K_4$, and *Loss Bin* as the branch one loss, branch two loss, branch three loss, branch four loss, and binary loss from the final outputs. The objective function of the whole network can be Equation (5) as:

$$Loss = Loss\ K_1 + Loss\ K_2 + Loss\ K_3 + Loss\ K_4 + Loss\ Bin, \tag{5}$$

in each branch, the loss function can be back-propagated from the sub-network to the task-specific network. Accordingly, the networks can be updated based on the four classification tasks. Our design considers the three predictions in a generalized framework, which is more suitable for the back-propagation of our multi-task deep neural network. The *loss* curve of first 40 epochs in training procedure is shown in Figure 8.

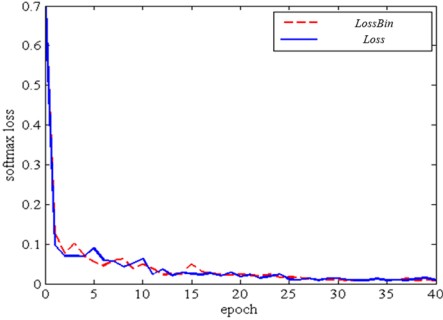

**Figure 8.** Loss curve with *Loss Bin* and *Loss*.

In summary, *loss* can be defined as Equation (6):

$$L(W) = \sum_{m=1}^{M} \sum_{i \in S^m} \alpha_m l^m (X_i Y_i | W), \tag{6}$$

where $M$ is the number of detection branches and $\alpha$ is the weight of the corresponding *loss*. The corresponding *loss* of each detection layer is defined as Equations (7) and (8):

$$L(X, Y|W) = L_{cls}(p(X), y) + \lambda[y \geq 1]L_{loc}(b, \hat{b}), \tag{7}$$

$$L_{loc}(b, \hat{b}) = \frac{1}{4} \sum_{j \in \{x,y,w,h\}} smooth_{L_1}(b_j, \hat{b}_j), \tag{8}$$

where the optimal parameter can be defined as in Equation (9):

$$W^* = argmin_W L(W), \tag{9}$$

For each detection layer $m$, there is a training sample $Sm = \{Sm+, Sm-\}$. For a single image, the distribution of an image with an object and one without an object is unbalanced; therefore, we modify the sampling step to eliminate this imbalance via Equation (10). There are three sampling strategies: random, bootstrapping, and mixture.

$$L_{cls} = \frac{1}{1+\gamma}\frac{1}{|S_+|}\sum_{i \in S_+} -log p_{y_i}(X_i) + \frac{\gamma}{1+\gamma}\frac{1}{|S_-|}\sum_{i \in S_-} -log p_0(X_i). \tag{10}$$

### 3.4. Comparison to other Segmentation Networks

In this paper, a multiscale input CNN network is combined with preprocessed datasets to design a network with stronger learning ability and higher accuracy. The network is based on image multiscale patching, where 80% of the image patch is used for training and 20% of the image patch is used for verification. The training is done an epoch at a time, while the verification set is evaluated as a whole. The epoch lasts 100 iterations.

The multiscale segmentation network is compared to the original PixelNet and Unet networks. The three networks are based on the same database. This paper compares the Se, Sp, and Acc indicators, and the results are reflected in the table of results. Our network results are shown in the Figure 9. Note that we are here to obtain the accurate segmentation effect of the various feature structures of the fundus image.

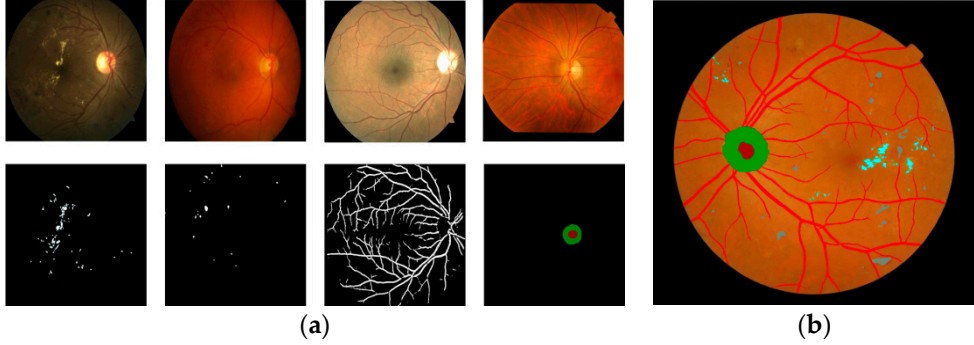

**Figure 9.** We show the segmentation of fundus images after training (from our own database). (**a**) shows the segmentation of exudates, roth spots, blood vessels, and optic discs (from left to right). (**b**) shows the segmentation of retinal structures and lesions.

This section compares the performance of our proposed method to related methods to demonstrate that the proposed multi-seg-CNN framework can indeed achieve better segmentation of retinal anatomy and pathological structure. Particularly, the proposed method is compared to three related state-of-the-art methods. The first is PixelNet (Aayush Bansal et al., 2017). The other is the classification baseline using the most widely accepted deep model, i.e., Unet (Olaf Ronneberger et al., 2015). The last

method is v3+ (Liang-Chieh Chen et al., 2018), which is Google's recent work on the segmentation. Our baseline model trains over three independent classification models using our own database for training. We then derive the segmentation method according to the clinical segmentation criteria. Figure 10 provides the ROC curve of each framework. All four methods use deep learning methods to train models using the same training data.

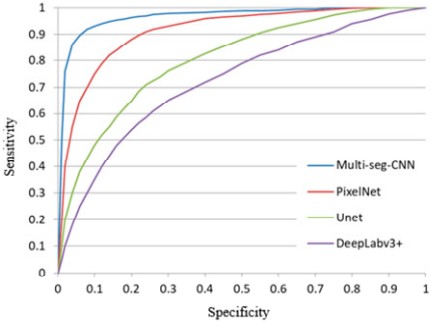

**Figure 10.** The ROC curve of different frameworks.

The best segmentation threshold and the highest accuracy during testing is achieved for images from a single database. Images from this database also have the fewest number of false positives and false negatives. Our method has high segmentation accuracy. In second place is the PixelNet method, which also has high segmentation accuracy, but because it is a single branch, the accuracy is not as high as the four-branch method. The Unet segmentation effect is not ideal when blood vessels are separately segmented. The Deeplabv3+ segmentation method has the lowest accuracy, which leaves us stunned because DeepLabv3+ is generally effective for small targets where the overall structure is segmented.

We will evaluate the Se, Sp, and Acc indicators for each feature structure, separately, for blood vessels, exudates, bleeding points, visual cup optic discs, and macula. Since the structures of the optic disc and the macula are relatively simple, and the features are obvious, this paper considers the two structures together.

For the concentrated structure, the segmentation results of the other three network segmentation methods (for optic discs and the macula) are listed in Table 2.

**Table 2.** The results of the different networks of cup optic disc and macula.

| Index | Se | Sp | Acc |
|---|---|---|---|
| Database labeling | 0.9386 | 0.9978 | 0.9982 |
| Multi-seg-CNN | 0.9130 | 0.9940 | 0.9973 |
| PixelNet | 0.9134 | 0.9934 | 0.9974 |
| Unet | 0.8894 | 0.9870 | 0.9954 |

Comparing the three networks, we notice that the PixelNet accuracy is higher than that of Unet and our designed Mult-CNN; however, we also find that Multi-seg-CNN is better than Unet, which has similar accuracy to that of PixelNet. This is likely due to the fact that the optic disc and the macula are simple in structure, and the segmentation is relatively easy to identify, so the sensitivity and specificity of the three networks are not much different.

We used similar methods to find the overlay structure and to obtain the blood vessel segmentation results in Table 3, the exudate segmentation results in Table 4, and the bleeding point segmentation results in Table 5.

**Table 3.** Vessel segmentation accuracy by different networks.

| Index | Se | Sp | Acc |
|---|---|---|---|
| Database labeling | 0.8936 | 0.9278 | 0.9282 |
| Multi-seg-CNN | 0.9130 | 0.9240 | 0.9273 |
| PixelNet | 0.7940 | 0.8344 | 0.8456 |
| Unet | 0.8394 | 0.8970 | 0.8933 |
| DeepLabv3+ | 0.7826 | 0.8332 | 0.8421 |

**Table 4.** Exudate segmentation accuracy by different networks.

| Index | Se | Sp | Acc |
|---|---|---|---|
| Database labeling | 0.8986 | 0.9378 | 0.9352 |
| Multi-seg-CNN | 0.9030 | 0.9340 | 0.9386 |
| PixelNet | 0.8234 | 0.8434 | 0.8574 |
| Unet | 0.8354 | 0.8873 | 0.9004 |
| DeepLabv3+ | 0.8125 | 0.8336 | 0.8533 |

**Table 5.** Roth spots segmentation accuracy by different networks.

| Index | Se | Sp | Acc |
|---|---|---|---|
| Database labeling | 0.8736 | 0.9159 | 0.9188 |
| Multi-seg-CNN | 0.8835 | 0.9140 | 0.9043 |
| PixelNet | 0.7854 | 0.8334 | 0.8455 |
| Unet | 0.8594 | 0.8860 | 0.8953 |
| DeepLabv3+ | 0.7432 | 0.8128 | 0.8356 |

We find that the accuracy of the overall structure is lower than that of the centralized structure. The reason for this result is that the coverage structure of a pixel is much larger than its centralized structure. The three indicators Se, Sp, and Acc have no centralized structure with high accuracy, yet this result is unavoidable. Thus, the accuracy of the overall structure of the network vessels, exudates, and bleeding points is improved compared to other networks used for medical image processing. We also compare the DeepLabv3+, which represents the semantic segmentation network. We come to the conclusion that DeepLabv3+ has no obvious effect on the segmentation of the fundus or on finding features. DeepLabv3+ is not as good as PixelNet when considering the segmentation effect. Intuitively, Table 6 combines the results from Tables 2–5. In 2015, Olaf Ronneberger et al. [33] show that such a network can be trained end-to-end from very few images and outperforms the prior best method (a sliding-window convolutional network) on the ISBI challenge for segmentation of neuronal structures in electron microscopic stacks. Aayush Bansal et al. [20] explore design principles for general pixel-level prediction problems, from low-level edge detection to midlevel surface normal estimation to high-level semantic segmentation. In 2018, Liang-Chieh Chen et al. [34] extended DeepLabv3 by adding a simple yet effective decoder module to refine the segmentation results especially along object boundaries. All tables contain data from three open source networks and the new network we designed. Finally, we calculate the total result of the four networks in terms of the retinal anatomy and pathological structure. Multiscale segmentation achieves a more accurate segmentation effect.

**Table 6.** Retinal image segmentation accuracy by different networks.

| Methods | Unet | | PixelNet | | DeepLabv3+ | | Multi-Seg-CNN | |
|---|---|---|---|---|---|---|---|---|
| Severity Levels | Sensitivity | Specificity | Sensitivity | Specificity | Sensitivity | Specificity | Sensitivity | Specificity |
| Cup optic disc | 0.892 | 0.991 | 0.913 | 0.992 | - | - | 0.915 | 0.994 |
| Macula | 0.896 | 0.992 | 0.917 | 0.994 | - | - | 0.918 | 0.997 |
| Vessel | 0.841 | 0.901 | 0.794 | 0.835 | 0.782 | 0.835 | 0.914 | 0.921 |
| Exudate | 0.843 | 0.892 | 0.828 | 0.842 | 0.813 | 0.831 | 0.907 | 0.932 |
| Roth spots | 0.866 | 0.897 | 0.793 | 0.837 | 0.741 | 0.816 | 0.882 | 0.917 |
| Total result | 0.864 | 0.932 | 0.844 | 0.896 | 0.777 | 0.823 | 0.902 | 0.948 |

## 4. Conclusions

In this work, we propose a deep learning method to segment a retinal fundus image based on its anatomy and pathological structure. We base our diagnosis method on the fundus dataset we designed. Combining multiscale structural methods yields higher accuracy than single-branch networks. Our method achieves higher precision than current state-of-the-art methods. We also diagnose retinal disease from a small amount of fundus data with inaccurate labels via retinal anatomy and pathological structure segmentation that classifies blood vessels, exudates, cup optic disc, and roth spots in an image. However, deep learning algorithms require large-scale datasets and a dataset consisting of 1500 images is not enough for highly accurate results. By enlarging the dataset and combining the methods proposed in this paper, a more optimized model is expected in future.

Next, we discuss two open problems that we will pursue in future research. In order to judge fundus lesions, we will add text when training the model (e.g., text from, a doctor's pathology report). The introduction of text can improve our clinically oriented framework and improve the diagnostic capabilities of the method. We will use image translation to increase the image database. We can use Generative Adversarial Networks (GAN) to achieve image translation.

**Author Contributions:** L.G. was responsible for proposing the end-to-end deep learning image compression framework based on Multiscale Segmentation Network. H.C. performed the numerical simulations and wrote the paper. Z.X. and Y.L. gave some suggestions on the mathematical model and formula derivation.

**Funding:** This work was supported by the National Natural Science Foundation of China under grant No. 61771340, Tianjin Science and Technology Major Projects and Engineering under grant No. 17ZXHLSY00040, No. 17ZXSCSY00060 and No. 17ZXSCSY00090, the Program for Innovative Research Team in University of Tianjin (No. TD13-5034).

**Conflicts of Interest:** All authors declare no competing interests.

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
