# Peer review of "Extracting Retinal Anatomy and Pathological Structure Using Multiscale Segmentation"

_applsci, doi:10.3390/app9183669_

Round 1

Reviewer 1 Report

Comments formulated during my review are presented below. These are as follows:

1) As the image segmentation is the significant technique in this paper, the authors should briefly provide several groups of image segmentation methods used in medical imaging, namely:

*superpixel segmentation methods [a,b]

*watershed segmentation methods [c,d]

*active contour methods [e,f,g]

[a] "Superpixel-based segmentation for 3D prostate MR images". IEEE transactions on medical imaging, 2016, vol. 35(3), 791-801.

[b] "Superpixel and multi-atlas based fusion entropic model for the segmentation of X-ray images". Medical image analysis, 2018, vol. 48, 58-74.

[c] "Watershed segmentation for breast tumor in 2-D sonography". Ultrasound in medicine & biology, 2004, vol. 30(5), 625-632.

[d] "Automatic liver segmentation in MRI images using an iterative watershed algorithm and artificial neural network". Biomedical Signal Processing and Control, 2012, vo. 7(5), 429-437.

[e]  "Malignant and Benign Mass Segmentation in Mammograms Using Active Contour Methods". Symmetry, 2017, vol. 9(11),  Article Number: 277, MDPI

[f] "Semi–Automatic Corpus Callosum Segmentation and 3D Visualization Using Active Contour Methods". Symmetry, 2018, 10(11), 589.

[g]  "Automated Vessel Segmentation Using Infinite Perimeter Active Contour Model with Hybrid Region Information with  Application to Retinal Images". IEEE Trans. Med. Imaging, 2015, 34(9), 1797-1807. 

The above segmentation methods offer some alternatives to machine learning based segmentation.

2) In a separate paragraph it is required to provide some including remarks to further discuss the proposed methods, for example, what are the main advantages and limitations in comparison with existing methods?

3) Please give a frank account of the strengths and weaknesses of the proposed research method. This should include theoretical comparison to other approaches in the field. 

4) Page 9: please correct equations (9) and (10)

5) The authors need to present and discuss several solid future research directions. 

6) Finally, the paper needs to be revised for language.

Author Response

Dear Editors and Reviewers ,

Thank you very much for the advice given by the editors and reviewers. The paper has been proof-read by an English native and we have uploaded CERTIFICATE OF LANGUAGE EDITING by Analee M. Miranda, PhD. We have revised the paper by using the "Track Changes" function in Microsoft Word, so that changes are easily visible to the editors and reviewers.

Thank you very much for your consideration, and looking forward to hearing from you soon.

Best regards,

Sincerely,

Hengyi Che

Reviewer 2 Report

The paper explores an interesting and challenging medical imaging problem; fundus image segmentation, and proposes a multi-path multi-scale segmentation architecture, which is mainly based on PixelNet by replacing VGG16 with VGG19. Results have been reported on a curated (authors' own) dataset. - The novelty of the paper is seen medium to low and limited to adapting an existing architecture to a medical problem. - The manuscript is very hard to read due to grammatical mistakes and serious flow issues that reduces clarity. Authors are encouraged to have a scientific English writer to review the manuscript. - The argument about the "black-box" of deep learning is not clear; how is it only providing a single feature? - It is not clear if the fully connected layers predict the label of a single pixel or the segmentation of the whole image or image patch. - With fully connected layers, the number of trainable parameters increases significantly, increasing the chance of overfitting. Is it the case with the proposed architecture? what changes to be made if the architecture is to be converted to fully convolutional? how this would affect results especially in low-sample size scenarios? - The multi-scale aspect is not clear. Is each scale a downsampled version of the same image or a different receptive field around the optic nerve? - the loss function used for network training is not clearly stated and justified.

Author Response

(The authors gave the same response as above.)

Round 2

Reviewer 1 Report

Authors improved the article in this revision significantly. One minor thing:

- Please provide references to the methods used in Table 6, i.e. Unet []
PixelNet[], DeepLabv3+[], Multi-seg-CNN[]

Author Response

Dear Editors and Reviewers ,

Thank you very much for the advice given by the editors and reviewers. We have revised the paper by using the "Track Changes" function in Microsoft Word, so that changes are easily visible to the editors and reviewers.

Thank you very much for your consideration, and looking forward to hearing from you soon.

Best regards,

< Lei Geng, Hengyi Che, Zhitao Xiao, Yanbei Liu> et al.

Reviewer 2 Report

Here are my comments pertaining to the authors' responses.

Response 1: Fig 1 does not illustrate the four input branches, which is the main contribution of the paper. I recommend that authors conduct an ablation experiment where they quantify the effect of each input branch to the generalization performance.

Response 5: The argument that fully convolutional architecture cause loss of details (compared to fully connected) is not convincing as with fully connected layers, spatial information is totally lost. Please provide a more compelling argument supported by some results. Also, data augmentation should be simple enough to try out if limited data is an issue, which supports my point of overfitting in case of fully connected architecture.

Response 6: How this multiscale strategy different from the pooling that is done inside the network? with pooling layers, the network inherently provide features at multiscale, what is the advantages (conceptually and empirically) of feeding multiscale versions of the same image?

Author Response

(The authors gave the same response as above.)

Round 3

Reviewer 2 Report

Authors has responded/addressed my concerns.